

# Competition and fixation of cohorts of adaptive mutations under Fisher geometrical model

Jorge A. Moura de Sousa[1], João Alpedrinha[1], Paulo R.A. Campos[2] and Isabel Gordo[1]

[1] Instituto Gulbenkian de Ciência, Oeiras, Portugal
[2] Departamento de Fisica, Cidade Universitária, Universidade Federal de Pernambuco, Recife, Pernambuco, Brazil

## ABSTRACT

One of the simplest models of adaptation to a new environment is Fisher's Geometric Model (FGM), in which populations move on a multidimensional landscape defined by the traits under selection. The predictions of this model have been found to be consistent with current observations of patterns of fitness increase in experimentally evolved populations. Recent studies investigated the dynamics of allele frequency change along adaptation of microbes to simple laboratory conditions and unveiled a dramatic pattern of competition between cohorts of mutations, i.e., multiple mutations simultaneously segregating and ultimately reaching fixation. Here, using simulations, we study the dynamics of phenotypic and genetic change as asexual populations under clonal interference climb a Fisherian landscape, and ask about the conditions under which FGM can display the simultaneous increase and fixation of multiple mutations—mutation cohorts—along the adaptive walk. We find that FGM under clonal interference, and with varying levels of pleiotropy, can reproduce the experimentally observed competition between different cohorts of mutations, some of which have a high probability of fixation along the adaptive walk. Overall, our results show that the surprising dynamics of mutation cohorts recently observed during experimental adaptation of microbial populations can be expected under one of the oldest and simplest theoretical models of adaptation—FGM.

## INTRODUCTION

Understanding the mechanisms and dynamics underneath the adaptive process is still a great challenge in evolutionary biology. Even in relatively simple environments, evolution experiments demonstrate that this process often involves complex dynamics such as: (1) competition between clones carrying different adaptive alleles (*Desai & Fisher, 2007*; *Perfeito et al., 2007*; *Maharjan & Ferenci, 2015*); (2) hitchhiking, along with beneficial alleles, of neutral and even deleterious mutations (*Gerrish & Lenski, 1998*; *Desai & Fisher, 2007*; *Perfeito et al., 2007*; *Lang et al., 2013*; *Maharjan & Ferenci, 2015*); (3) second-order selection of mutations which lead to increased mutation rates and mutator phenotypes (*Sniegowski, Gerrish & Lenski, 1997*; *Tenaillon et al., 2001*; *Desai, Fisher & Murray, 2007*; *Perfeito et al., 2007*; *Barrick et al., 2009*; *Wielgoss et al., 2013*; *Maharjan & Ferenci, 2015*);

Corresponding author
Jorge A. Moura de Sousa,
jasousa@igc.gulbenkian.pt

or (4) the emergence of negative frequency-dependent interactions between genotypes (*Gerrish & Lenski, 1998*; *Maharjan, 2006*; *Desai, Fisher & Murray, 2007*; *Desai & Fisher, 2007*; *Perfeito et al., 2007*; *Herron & Doebeli, 2013*; *Lang et al., 2013*; *Maharjan & Ferenci, 2015*). It is increasingly evident that not only these dynamics influence the adaptive process but also that they emerge as a result of the adaptive process. For instance, the fixation of mutator phenotypes has been typically observed in adapting populations, as their higher mutation rate provides them with a higher probability of acquiring and hitchhiking with rare beneficial mutations (*Chao & Cox, 1983*; *Taddei et al., 1997*; *Tanaka, Bergstrom & Levin, 2003*; *Desai & Fisher, 2007*; *Gentile et al., 2011*; *Torres-Barceló et al., 2013*; *Lang et al., 2013*). More recently, experimental findings from microbial evolution experiments coupled with sequencing analysis unveiled that a dramatic level of polymorphism in populations can occur during adaptation (*Lang et al., 2013*; *Frenkel, Good & Desai, 2014*; *Maddamsetti, Lenski & Barrick, 2015*). Interestingly, mutation cohorts, consisting of multiple mutations that segregate and reach fixation simultaneously, are observed in populations adapting to simple environmental laboratory conditions. In large populations, the input of new mutations can be so high that new mutants emerge in backgrounds already carrying other mutations, leading to the formation and competition between mutation cohorts. Such competition results in longer times for mutations to reach fixation and complex dynamics, as different mutations aggregate in separate groups. Indeed, synchronous increase or decrease in frequency of these mutations, competition between distinct cohorts and the simultaneous fixation of the mutations that form the cohorts is a pervasive observation during this laboratory microbial adaptations (*Sniegowski, Gerrish & Lenski, 1997*; *Tenaillon et al., 2001*; *Barrick et al., 2009*; *Wielgoss et al., 2013*; *Lee & Marx, 2013*; *Lang et al., 2013*; *Maddamsetti, Lenski & Barrick, 2015*).

A classical model of adaptation to a novel environment, theorized almost 100 years ago by Fisher before the structure of DNA was discovered, is Fisher's Geometrical Model (FGM). It is a simple model where a population adapts towards a fixed phenotypic optimum (*Fisher, 1930*). FGM considers the process of adaptation assuming that individuals are defined by their traits under selection, which are geometrically represented in a defined multidimensional landscape. In this model, directionality in selection emerges by assuming that fitness is related to the distance of each phenotype to the optimum. Thus, a population moves towards the fitness peak through the gradual accumulation of beneficial mutations. FGM has been extensively studied beyond its original scope to make predictions under different scenarios about the distribution of beneficial mutations during adaptation (*Orr, 1998*; *Martin & Lenormand, 2008*; *Bataillon, Zhang & Kassen, 2011*), the level of epistasis between mutations (*Martin, Elena & Lenormand, 2007*; *Blanquart et al., 2014*), the effects of deleterious mutations accumulated under relaxed selection (*Martin & Lenormand, 2006*; *Perfeito et al., 2014*), the effect of drift load in the fitness at equilibrium (*Otto & Orive, 1995*; *Lourenço, Galtier & Glémin, 2011*), sympatric speciation in an environment with multiple fitness peaks (*Barton, 2001*; *Sellis et al., 2011*) and the effect of mutation pleiotropy (the number of traits affected by a single mutation) in adaptation (*Welch & Waxman, 2003*; *Chevin, Martin & Lenormand, 2010*; *Lourenço, Galtier & Glémin, 2011*). *Martin (2014)* recently proposed that FGM basic assumptions can emerge from models

which consider the nature of complex metabolic networks within a cell. FGM predictions are largely compatible with observations coming from experimental evolution studies, mostly in microorganisms (*MacLean, Perron & Gardner, 2010*; *Chou et al., 2011*; *Khan et al., 2011*; *Sousa, Magalhaes & Gordo, 2012*; *Gordo & Campos, 2013*; *Weinreich & Knies, 2013*; *Tenaillon, 2014*).

Here we ask whether the patterns of competition and fixation of simultaneous segregating mutations (mutation cohorts) along an adaptive walk observed experimentally can be reproduced under FGM. We study FGM under clonal interference by simulating populations with a large mutational input ($NU \gg 1$, where $N$ is the population size and $U$ the genomic mutation rate), where both beneficial and deleterious mutations occur, therefore generating competing polymorphisms (*Gordo & Campos, 2013*). Since the simplest version of FGM assumes full mutational pleiotropy, which is a restrictive assumption and thought to bear poor biological realism (*Welch & Waxman, 2003*; *Orr, 2005*; *Chevin, Lande & Mace, 2010*; *Wang, Liao & Zhang, 2010*; *Wagner & Zhang, 2011*; *Lourenço, Galtier & Glémin, 2011*), we also studied a model assuming partial pleiotropy. The degree of mutational pleiotropy is expected to influence the dynamics of adaptation (*Wagner & Zhang, 2011*). In our model of partial pleiotropy, similar to that of *Lourenço, Galtier & Glémin (2011)*, a single mutation can only change a subset of traits ($m$), taken at random from the full set of traits ($n$) that contribute to fitness. When populations have small sizes or mutation rates are low, the analytical expressions for predicting the rate of adaptation under this model suggest that mutational pleiotropy can affect the dynamics of adaptation of populations approaching a fitness peak (*Lourenço, Galtier & Glémin, 2011*). However, such analytical results rely on a strong simplifying assumption: the populations are monomorphic most of the time. This assumption is quite restrictive given the increasing experimental evidence for high rates of beneficial mutations both in natural (*Eyre-Walker & Keightley, 2007*; *Jensen, Thornton & Andolfatto, 2008*) and in experimental populations (*Perfeito et al., 2007*; *Good et al., 2012*), which promptly produce competition between segregating mutations arising in distinct lineages and drive the dynamics of mutation cohorts described above. To address these more relevant scenarios, we use stochastic simulations of FGM for populations undergoing strong clonal interference. We consider large populations and values of mutation rate and mean effect of mutations that are in reasonable agreement with current estimates for microbial populations (*Gordo, Perfeito & Sousa, 2011*; *Perfeito et al., 2014*).

Most of theoretical analysis done so far focused on predicting the equilibrium mean fitness, and did not address the time scale at which such equilibrium is in fact reached. As experiments where evolution is followed for longer and longer periods are emerging (*Lang et al., 2013*; *Barrick & Lenski, 2013*), it is also important to have theoretical expectations on the full dynamics of the approach to equilibrium under classical models of adaptation, both at the phenotypic and genotypic level, as we do here. By tracking each individual mutation during the adaptive walk as the populations approach the optimum, we find that the simplest version of FGM can generate the complex mutation cohort dynamics observed in microbial adaptation experiments, under specific evolutionary parameters within a biological realistic range.

## METHODS

### Simulation methods of fisher geometrical model

FGM considers each individual as a point in a $n$-dimensional space, where $n$ is the number of traits under selection. Each individual is characterized by a vector of coordinates $(z_1, z_2, \ldots, z_n)$ that gives the position of the individual in the fitness landscape. This vector represents the phenotypic values for each trait. Without loss of generality, we define the optimum as the origin of the $n$-dimensional space. As commonly done, we assume that fitness is given by a Gaussian function of the distance to the optimum, $w = \exp(-\sum_{i=1}^{n} z_i^2)$. We assume that mutations, as rare events, follow a Poisson distribution with a genomic mutation rate $U$, per individual, per generation. Each mutation changes $m$ traits chosen at random from the total $n$ traits, and the effect it causes in each affected trait follows a normal distribution with mean 0 and variance $\sigma^2$. We consider a Wright–Fisher model and assume multinomial sampling with fixed population size $N$. The contribution of each individual to the next generation is proportional to its fitness and it is based on a multinomial sampling of the population. We assume large population sizes, as typical in microbial laboratory adaptation experiments (*Barrick & Lenski, 2013*), and consider values of genomic mutation rates $(U)$ that are reasonable for microbes. Indeed, the order of magnitude of the genomic mutation rate has been previously estimated for many species of organisms and falls within $U \sim 0.001$ (*Drake et al., 1998*; *Lee et al., 2012*). We explored values for the complexity within a range that is in accordance with estimates obtained from experiments using different organisms including viruses, bacteria and multicellular organisms (reviewed in *Lourenço, Galtier & Glémin (2011)*, specifically Fig. 2 of that paper). We also explored different values of the mean effect of mutations ($E(S) = -m\sigma^2$, under the assumptions of partial pleiotropy in FGM, described above) ranging from very small $-0.1\%$, to much larger ($-20\%$), as estimated from different mutation accumulation experiments in different organisms (*Martin & Lenormand, 2006*; *Eyre-Walker & Keightley, 2007*; *Gordo, Perfeito & Sousa, 2011*). The code for the simulations is provided as Files S1–S4.

## RESULTS

### Dynamics of approach to equilibrium mean fitness

We start by studying the dynamics of fitness increase along tens of thousands of generations for different levels of phenotypic complexity ($n$), pleiotropy ($m$) and mean effect mutations ($E(S)$). Figures 1A–1D shows that the initial rate of fitness increase is lower under low pleiotropy across all values of the mean fitness effect of mutations ($E(S)$) studied. The effect is particularly strong for $|E(S)| > 0.01$. However, in the long run, populations with lower pleiotropy reach higher levels of mean fitness (see also Fig. S1). Increasing complexity, while maintaining a similar level of pleiotropy, shows a similar pattern for the fitness plateau, where we find that populations with fewer traits reach higher fitness values within the simulated time period (Figs. 1E and 1F).

### Mutation cohorts of fixed in the initial steps of adaptation

Next, we studied the dynamics of mutation fixation along the adaptive walk. We first studied populations with maximum pleiotropy and various degrees of complexity across

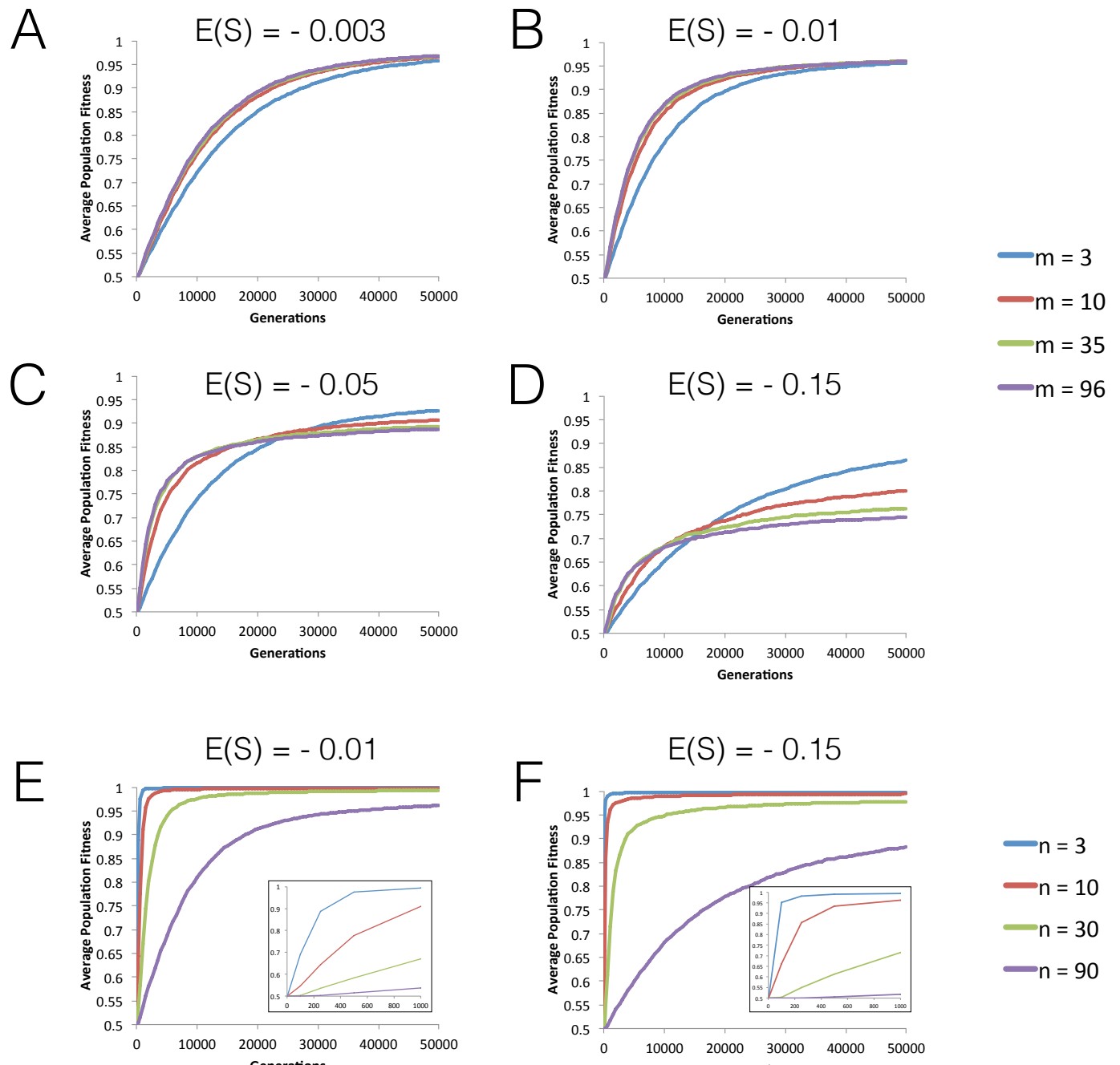

**Figure 1  Dynamics of mean fitness increase under FGM with partial pleiotropy.** (A–D) Dynamics of mean fitness of asexual populations with varying degrees of pleiotropy ($m$). All populations have high complexity ($n = 96$) and distributions of fitness effects (DFEs) with different means are studied. Other parameters are population size $N = 10^4$, mutation rate $U = 0.001$, initial fitness $w_0 = 0.5$. The variance for the mutation effects $\sigma^2$ varies as $m$ varies, so that $E(S)$ (which is $-m\sigma^2$ in this model) has the value indicated in each panel. (E–F) The effect of increasing phenotypic complexity ($n$) on the dynamics of fitness increase. Other population parameters are: $N = 10^4$, $U = 0.001$, variance $\sigma^2 = 0.004$, $m = 3$ and the initial fitness $w_0 = 0.5$. Short-term dynamics are highlighted as an inset.

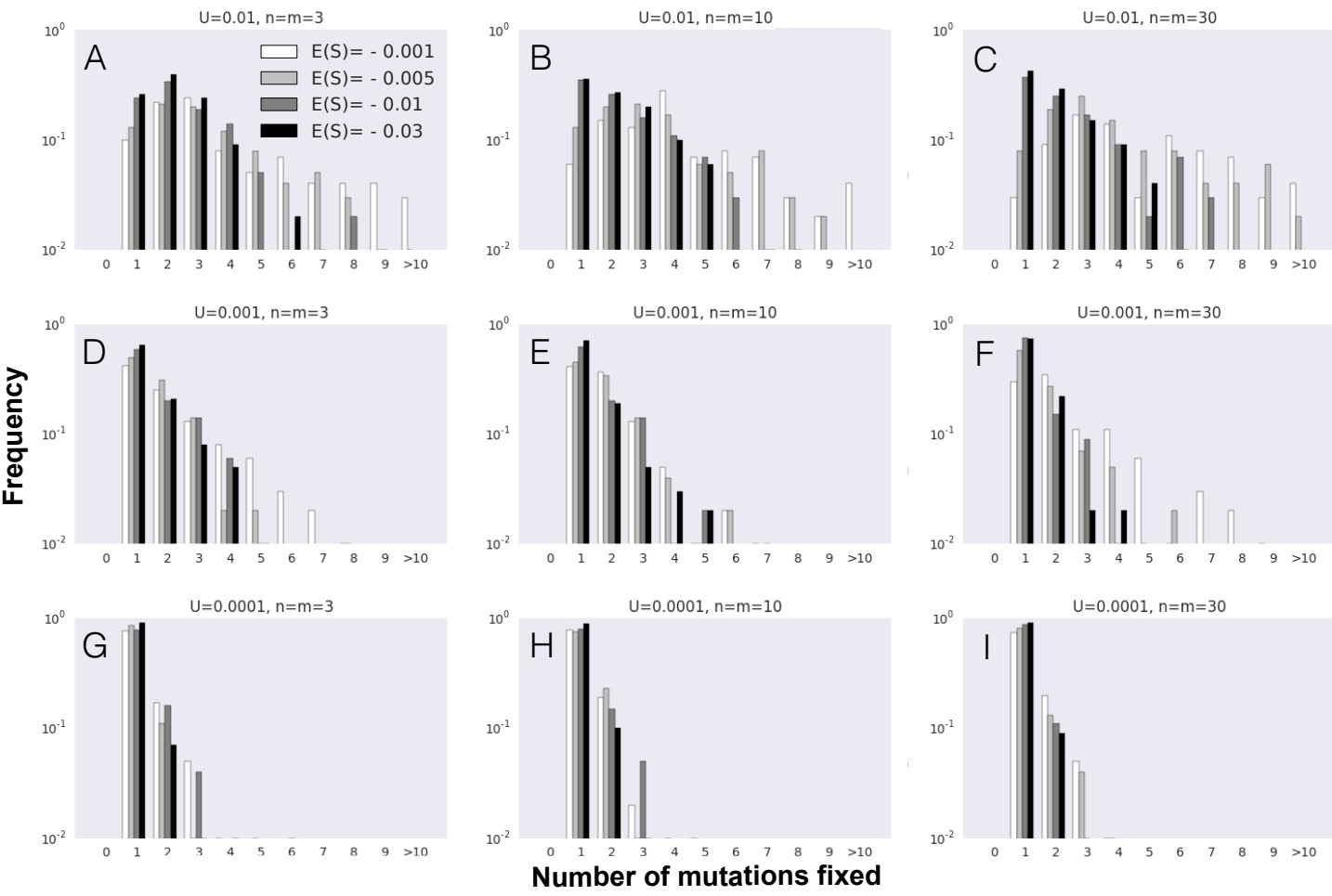

**Figure 2** **FGM can lead to simultaneous fixation of mutation cohorts.** (A–I) The probability distribution of the number of mutations fixed during the first fixation event in the adaptive walk, i.e., mutation cohort size. Parameter values used in these simulations are as follows: population size $N = 10^4$ and initial fitness $w_0 = 0.5$ are the same across all panels; the other parameters vary as indicated in each panel (mutation rate $U$ increases from 0.0001 (G–I) to 0.01 (A–C), implying higher levels of clonal interference; phenotypic complexity $n$ (and pleiotropy $m = n$) increase from (A), (D) and (G) to (C), (F) and (I); and, within each panel, $s^2$ varies, implying different distributions of arising mutations with mean selective effects of $E(S)$ indicated by the different shades). Data is shown for 100 simulations per combination of parameters.

different $E(S)$ and mutation rates, and asked how many mutations fix simultaneously in the first step, i.e., the mutation cohort size at the first fixation step. Figure 2 shows that fixations of cohorts of mutations can be very common, reflecting the degree of clonal interference occurring in these large populations. Across all parameters, the major determinant of the number of mutations fixing in cohorts is the mean effect of mutations $(E(S))$, with lower effect mutations promoting fixation of cohorts of larger size. The other relevant parameter to the size of the fixed cohorts is, as expected, the mutation rate, with an increased mutation rate showing the largest cohorts of mutations fixed. Therefore, the combination of small effect mutations generated at a high rate leads to the fixation of larger mutation cohorts. We performed the same analysis on simulations where we relax the assumption of full pleiotropy. Populations with partial pleiotropy ($m = 3$, 10 or 20) for the highest level of complexity previously tested ($n = 30$) show patterns that are qualitatively similar (Fig. S2).
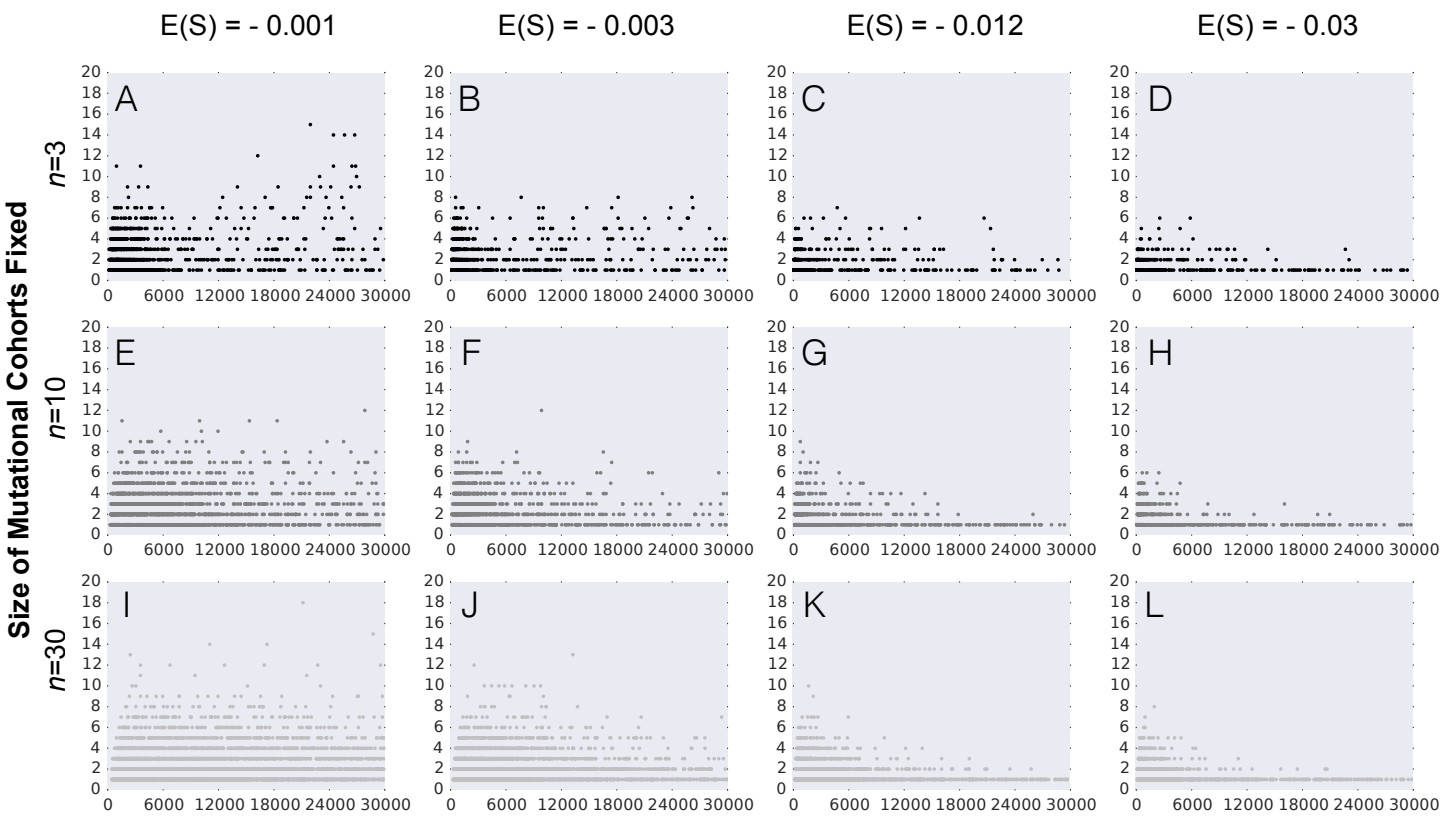

**Figure 3** **Number of mutations fixed (mutation cohort size) along the adaptive walk.** (A–L) Parameter values are population size $N = 10^4$, the initial fitness $w_0 = 0.5$, mutation rate $U = 0.001$ and other parameters as indicated in each panel (with phenotypic complexity $n$ taking the values indicated to the left of panels A, E and I, and mean effect of mutations $E(S)$ taking the values indicated above panels A, C, D and E). In all panels full pleiotropy is assumed ($m = n$). Data is shown for 100 simulations per combination of parameters.

The main difference detected occurs in simulations with a high mutation rate, where the likelihood of observing large stronger effect mutation cohorts increases relative to the case of full pleiotropy. Additionally, both in the cases of full or partial pleiotropy, the complexity of evolving populations shows a minimal effect on the size of the fixed mutation cohorts. Therefore, the number of mutations observed fixing simultaneously in the first step of adaptation is mainly determined by the mutation rate and the mean selective effect of mutations.

## Mutation cohorts fixed along the adaptive walk

In order to understand how the probability of observing the fixation of mutation cohorts changes along the adaptive walk, we next study the distribution of mutations fixed beyond the first step of adaptation. Figure 3 shows the pattern of mutation cohorts fixed along an adaptive walk lasting 30,000 generations. Each point in the panels of Fig. 3 corresponds to a fixation event occurring during this time period, with the number of mutations (i.e., the size of the cohorts) that compose each of these fixations represented in the $y$-axis. The probability of observing cohorts consisting of a large number of mutations later in

the adaptive walk is strongly dependent on the average selective effect of the mutations (contrast in Figs. 3A, 3E and 3I with 3D, 3H and 3L ). Lower effect mutations lead to the fixation of cohorts of larger sizes not only in the first steps, but also as populations approach the equilibrium fitness. Interestingly, we observe that, for the lower values of mean mutation effect, the likelihood of fixing mutation cohorts of larger sizes (from 4 to 8 mutations) increases for populations with a higher complexity, throughout the adaptive walk. For high values of $|E(S)|$ and populations with a lower number of traits, fixation of large mutation cohorts becomes an increasingly rare event once they approach the fitness equilibrium.

Overall, along the adaptive walk, the size of fixed mutation cohorts tends to shrink, at a faster pace for large values of $|E(S)|$. These simulations therefore suggest that, for long-term adaptation of populations approaching a fixed optimum, fixation of single mutations is expected to become the dominant pattern. However, when $|E(S)|$ is small (panels in Fig. 3, where $|E(S)|$ is 0.001 or 0.003) that regime may take a substantial time to be reached.

We have also explored the role of partial pleiotropy on the size of mutation cohorts fixed along the adaptive walk. The results are qualitatively similar to the ones observed for populations with full pleiotropy, with the size of mutation cohorts generally decreasing as populations get closer to the fitness peak, but the decrease taking longer periods of time as $|E(S)|$ becomes smaller (see Fig. S3A). Furthermore, the distributions of cohort sizes of the mutations simultaneously fixed along the adaptive walks, for populations under partial and full pleiotropy, are similar, as shown in Fig. S3B. Overall, these observations indicate that partial pleiotropy plays a minor contribution to the fixation of multiple mutations.

## Dynamics of mutation cohorts

Finally, we study the dynamics of polymorphism expected in populations climbing the Fisherian landscape. We first focus our simulations on short-term evolution, a time scale for which polymorphism data has been obtained recently for yeast strains adapting to a simple laboratory environment (*Lang et al., 2013*). Figure 4 shows the dynamics of frequency change of each individual mutation segregating in populations evolving for 1,000 generations. Aggregation of cohorts of mutations can be clearly observed across the different replicate populations, all simulated with the same evolutionary parameters. The parameter set shown was chosen to be one where we could find a pattern similar to that observed in the evolution experiments done in yeast (*Lang et al., 2013*). In the replicate simulated populations, just as in the replicate experimental ones, cohorts of different sizes emerge and compete against each other, with some achieving fixation and others being outcompeted. Although this phenomenon of "cohort interference" is more likely for cohorts competing at lower frequencies (where many mutations are segregating), it can also be observed when mutations reach high frequencies (e.g., Figs. 4A and 4C). Even under the same parameter values different patterns can be observed among the replicates: sequential fixation of cohorts of low size in some populations (e.g., Figs. 4D and 4G) and fixation of cohorts of large size in other populations (e.g., Figs. 4E, 4F and 4I). The same qualitative behavior is observed when simulating a higher number of replicate populations adapting under FGM (see Fig. S4).
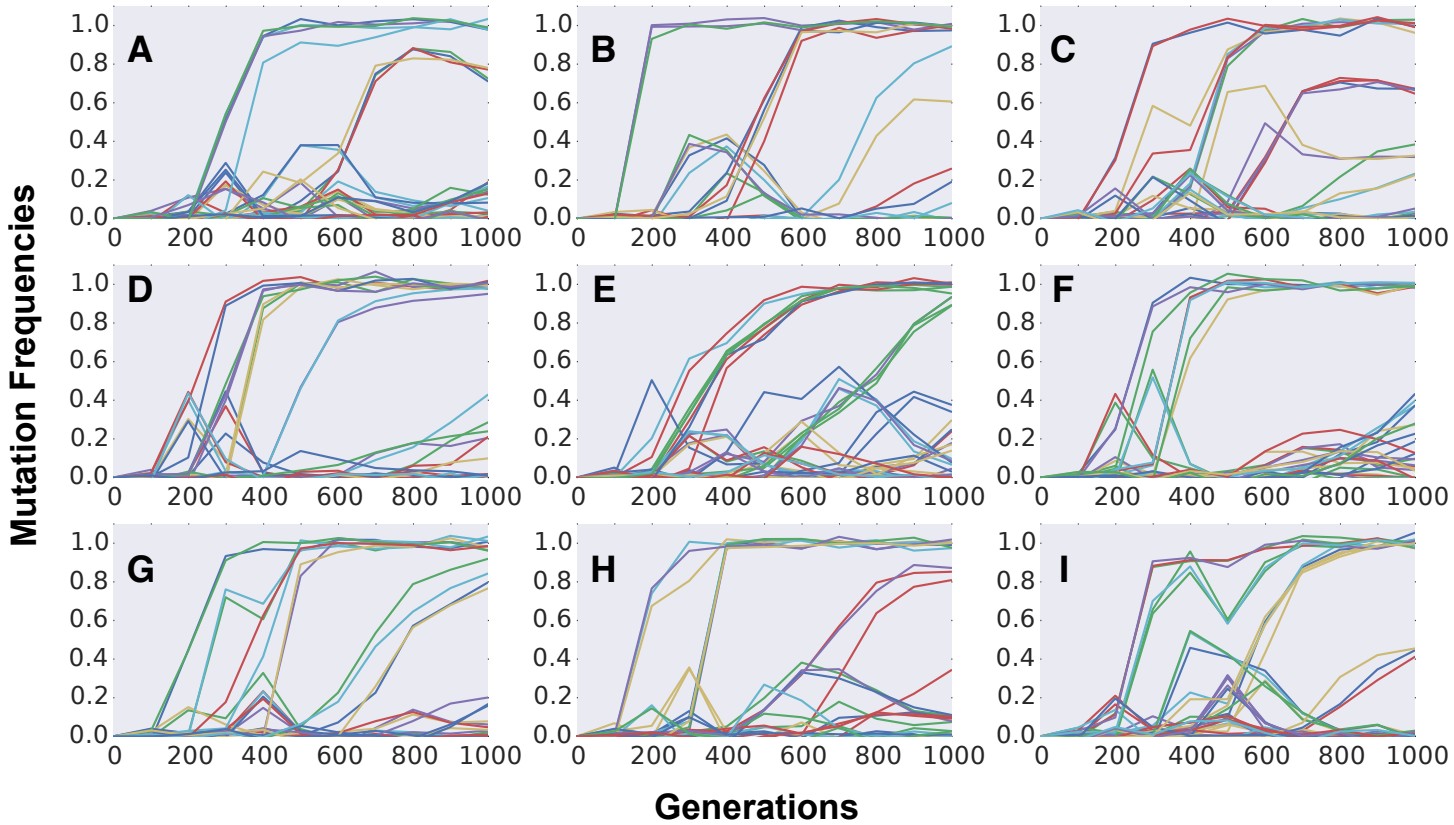

**Figure 4  Dynamics of frequency change of individual mutations along time across independently evolving populations.** (A–I) Aggregation of multiple mutations in cohorts can be easily detected by the simultaneous increase in frequency of different mutations (in different colors). Competition between cohorts can be commonly observed during the first 1,000 generations of adaptation. Parameter values used are population size $N = 10^5$, initial fitness $w_0 = 0.5$, mutation rate $U = 0.003$, $n = 10$, $m = 3$ and $E(S) = -0.012$. Gaussian noise (mean 0, variance 0.02), mimicking experimental error, was added to the dynamics for increasing visibility and comparison with experimental data.

The formation of mutation cohorts under FGM can be observed across several parameter sets, under different values of complexity and mean mutation effects (see Fig. S5 for examples). Yet some sets of parameters lead to results more consistent with the observations originating from evolution experiments than others. In the yeast evolving populations analyzed in *Lang et al. (2013)*, an average of 25 different mutations segregating and an average of 6 fixations were observed, across the replicate populations. Moreover, considerable fitness increases could be detected (around ∼6% after 500 generations). Across the sets of parameters we explored, this level of fitness increase and high number of mutations was unlikely when simulating FGM with high complexity ($n = 90$, rightmost panels in Figs. S5A and S5B). Moreover, out of 40 independently evolving populations in *Lang et al. (2013)*, in all but one replicate more than two mutation cohorts were observed to be segregating (population BYB1-B01). A pattern with few mutation cohorts was detected in the simulations sets we performed under full pleiotropy, low complexity and strong effect mutations (leftmost top panel in Fig. S5A).

Under the parameter values assumed in Fig. 4, we followed the dynamics of frequency change beyond 1,000 generations to ask about the extent to which polymorphisms would

be lost in the long run. Remarkably the simulations indicate that polymorphisms can be maintained for a very long time (sometimes more than 10,000 generations) (see Fig. S6). The simulations also indicate that the emergence of new mutation cohorts of large size becomes less frequent than that observed during the first 1,000 generations. Data from experiments involving longer time periods may thus help determining if the predictions of this model are met.

## DISCUSSION

With the advances of next generation sequencing, increased power to observe the dynamics of adaptation at the resolution of individual mutations has emerged. The data recently gathered indicates that adaptation of microbial populations adapting in laboratory environments exhibit patterns very distinct from the classic single selective sweep model of periodic selection. Instead the dynamics of molecular evolution in these microbes evolving in real time shows that aggregates of beneficial mutations segregate and fix simultaneously (*Lang et al., 2013*; *Maddamsetti, Lenski & Barrick, 2015*; *Zanini et al., 2016*). Even though Fisher's model is a phenotypic model of adaptation, the easiness by which this simple non-gene centric model is able to produce dynamics of fitness change similar to those observed in such experiments, lead us to ask if such dynamics of molecular change could be expected under this model. The simulations performed show that Fisher's Geometric Model, in its simplest version, can reproduce dynamics of cohort interference such as the ones observed in experimental settings. As observed for the frequencies of sequenced mutations in evolve and re-sequence experiments in yeast and bacteria, the mutation dynamics of simulated populations under FGM can be non-monotonic and exhibit patters of interference between clones belonging to distinct mutation cohorts. The number of mutations that compose these cohorts are found to be variable and the polymorphisms emerging can last for thousands of generations. We note that FGM does not consider social and ecological interactions that are likely to be important in explaining genetic diversity in natural populations (*Cordero & Polz, 2014*), nor does it consider frequency dependent selection, which has been shown to also occur in laboratory evolving microbial populations adapting to simple ecological conditions (*Maharjan et al., 2012*; *Herron & Doebeli, 2013*).

Remarkably, when simulating the dynamics of individual mutations produced under FGM, we could find parameter sets leading to patterns very similar to the ones that are increasingly being assayed through whole genome sequencing of evolving microbial populations (e.g., compare Fig. 4 with Fig. 1 in *Lang et al. (2013)*). Although such patterns are dependent on the parameters used (see Fig. S5), they could be observed in simulated populations assuming a set of parameters within a biological plausible range: a mean effect of mutations around 1%, consistent with measurements in microbes (*Kibota & Lynch, 1996*; *Zeyl & DeVisser, 2001*), and a genomic mutation rate of $3 \times 10^{-3}$, consistent with Drake's rule (*Drake et al., 1998*; *Lee et al., 2012*).

Under FGM, cohort interference can be common during the initial steps of adaptation, and is more likely when the mean effect of mutations is small and mutation rates are not too small. In these scenarios many small effect mutations simultaneously segregate, each taking
a long time to reach fixation, which likely results in the acquisition of additional mutations (either beneficial, neutral or slightly deleterious) in the same genetic background. In contrast, when $E(S)$ is large, beneficial mutations sweep to fixation faster, and the likelihood of acquiring additional mutations in their background diminishes. As expected the size of the interfering cohorts increases as the mutation rate increases, since an increased amount of mutations segregating in these high $U$ populations prevents the fast fixation of a single mutation.

Given that levels of complexity and pleiotropy may differ across genomes and environments, we further enquired if the patterns described above would change for populations where differences between complexity and pleiotropy are very large. We thus performed simulations with $n = 500$ and a low degree of pleiotropy ($m = 3$) for different values of $E(S)$. Adaptation in these scenarios occurs substantially slower, due to the very high dimensionality of the fitness landscape (Fig. S7A). The sizes of mutation cohorts initially fixed can also be large (when $|E(S)|$ is small), similarly to the simulations under lower complexity (Fig. S7B). However, fixations now involve long waiting times, often more than 2,000 generations (Fig. S7C). The simulations also indicate that fitness increase resulting from the fixation of mutation cohorts can be very small (Fig. S7D). Thus, both data of mutation frequency dynamics and of fitness increase along time are required to determine the levels of complexity of the fitness landscape.

The relationship between the size of cohorts and both the mean effect of mutations and the mutation rate is also detected when we study adaptation over longer periods (Fig. 3). The size of fixed cohorts tends to shrink along the adaptive walk, and does so at a faster pace for large values of $E(S)$. Therefore, for populations approaching a fixed optimum the pattern of long-term adaptation is expected to become dominated by fixation of single mutations. However if $E(S)$ is small such pattern may take many thousands of generations to be detected (right panel in Fig. 3), a time scale that is out of reach for most laboratory experiments so far studied. The famous long-term evolution experiment (LTEE) in *Escherichia coli* constitutes an important exception, where patterns of adaptation can be studied over periods as long as 60,000 generations (*Lenski et al., 1991*; *Maddamsetti, Lenski & Barrick, 2015*). The access to samples frozen every 500 generations allows the tracking of individual mutations and the reconstruction of the evolutionary genetic history of an individual population. *Maddamsetti, Lenski & Barrick (2015)* tracked the emergence of 42 mutations in one of the evolving populations and showed competition and interference between lineages carrying several mutations, including the simultaneous fixation of these sets. In this population however, not only clonal interference was observed but also frequency-dependent selection was important in driving the dynamics of mutation cohorts. On a shorter-term experiment also with *E. coli* but now evolving in a chemostat *Maharjan et al. (2015)* detected synchronous sweep of multiple mutations but the levels of polymorphism were also driven by frequency dependent interactions between clones. As we show here clonal interference alone can lead to dynamics of cohort interference, but given the emergence of frequency dependent selection even in the simplest environments, as well as its potential critical role in natural microbial populations (*Cordero & Polz, 2014*),

it will be important in future work to model other fitness landscapes which can allow for the simultaneous occurrence of both processes.

## CONCLUSIONS

In the current work, we study a simple version of the Fisher's Geometrical Model that assumes partial or full pleiotropy. Despite its simplicity, FGM has been successfully used to reproduce patterns of the dynamics of the adaptive process (*Chevin, Martin & Lenormand, 2010*; *Martin, 2014*). A common pattern emerging from the short-term dynamics of populations of microorganisms evolving in laboratory conditions is the finding that mutants carrying multiple segregating mutations can go to fixation (*Lang et al., 2013*; *Maddamsetti, Lenski & Barrick, 2015*). Before resorting to more complex models of fitness landscapes (*Cordero & Polz, 2014*), we inquired whether a simple and less parameterized model, such as FGM, could capture the essence of this sort of observation under reasonable parameters. Assuming large population sizes close to those in the experiments, and mutation rates typical of microbes, thus naturally driving population to a clonal interference regime, we show that FGM, both under full and partial pleiotropy, generates patterns of segregation and competition of cohorts of mutations that are consistent with experimental observations.

## ACKNOWLEDGEMENTS

IG thanks Olivier Tenaillon for pointing out the potential importance of partially pleiotropy leading to different patterns of mutation accumulation.

### Funding

The research received funding from the European Research Council under the European Community's Seventh Framework Programme (FP7/2007–2013)/ERC grant agreement no 260421–ECOADAPT and from Fundação para a Ciência e Tecnologia (FCT) through PFE-GI-FCT-PTDC/BIA-EVF/118075/2010. IG acknowledges the salary support of LAO/ITQB & FCT. JAMS acknowledges the salary support of FCT through the grant SFRH/BD/89151/2012. PRAC is partially supported by Conselho Nacional de Desenvolvimento Científico e Tecnologico (CNPq). The funders had no role in study design, data collection and analysis, decision to publish, or preparation of the manuscript.

### Grant Disclosures

The following grant information was disclosed by the authors:
The European Research Council: 260421–ECOADAPT.
Fundação para a Ciência e Tecnologia (FCT): PFE-GI-FCT-PTDC/BIA-EVF/118075/2010, SFRH/BD/89151/2012.
LAO/ITQB & FCT.
Conselho Nacional de Desenvolvimento Científico e Tecnologico (CNPq).

## Competing Interests

The authors declare there are no competing interests.

## Author Contributions

- Jorge A. Moura de Sousa, João Alpedrinha, Paulo R.A. Campos and Isabel Gordo conceived and designed the experiments, performed the experiments, analyzed the data, contributed reagents/materials/analysis tools, wrote the paper, prepared figures and/or tables, reviewed drafts of the paper.

## Data Availability

The code used has been supplied as a Supplemental Dataset.

## Supplemental Information

Supplemental information for this article can be found online at http://dx.doi.org/10.7717/peerj.2256#supplemental-information.

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
