# Peer review of "Competition and fixation of cohorts of adaptive mutations under Fisher geometrical model"

_PeerJ, doi:10.7717/peerj.2256_

## Round 0.1 · original submission · Minor Revisions

· Academic Editor

Minor Revisions

This is a very interesting manuscript that will appeal to a broad readership in evolution and ecology. The reviewers are very positive about the study and make constructive suggestions that will aid in presentation, as well as aid in more strongly connecting the interesting findings presented to experimental data.

I recommend that the authors submit a revised manuscript that addresses the reviewers' suggestions.

Reviewer 1 ·

Basic reporting

I have no problem with understanding English of the manuscript, but do not take my word for it as I am not an English native speaker. Also, ensuring there are no typos in the paper is traditionally the Editor’s duty, not mine.

The paper uses word ‘cohort’ for “multiple mutations spreading together” (see lines 68-69). Whereas ecology defines ‘a cohort’ as a group of individuals form the same population sharing a same characteristic (e.g. same age, similar size, a facts they are all carrying a parasite). I would like to encourage authors to find a different term, specially that they ask questions on the edge of evolution and ecology.

Figure 2 of the manuscript consist of 9 panels and can be made clearer by removing redundant legends and axis labels. The same refers to supplementary Figure S2 (see attached PDF with my comments).

Figures 3 and 4 are missing axis labels.

Experimental design

Experimental design is sound. The code of simulation is made available but unfortunately I was unable to run the simulation program. It compiled without errors:
g++ peerj-9188-SuppFile3_CodeFGMPleio-runTrackIndividualMutations.cp -o name -lgsl -lgslcblas
But when attempted to run (with parameters given in the comments of the source file):
./name 10000 0.5 96 3 0.001 2000 100 0 1 17847718 0.001
it thrown a segmentation fault on Ubuntu 14.04 STL, 64 bit system, g++ version 4.8.4. I did not try very hard to run in, though.
Please, if the software has any specific dependencies or relies in specific versions of libraries, state so in the comments of the code or in the documentation attached.

Validity of the findings

After quite impressive introduction I am bit disappointed in the article discussion section. There is discussion considering comparison with experimental findings in microbial genetics which shows that the simple FGM can, in fact, mimic current findings in microbial evolution. Which is good. Though, I am missing an explanation what does in mean for biology? Unlike many modern more complicated models, FGM in NOT a gene-centric model. How these finding link to current gene-centric view of evolution? Consider this review article:
Cordero, O. X. & Polz, M. F. Explaining microbial genomic diversity in light of evolutionary ecology. Nat. Rev. Microbiol. 12, 263–273 (2014).

Additional comments

Some minor remarks are in the annotated PDF.

Annotated reviews are not available for download in order to protect the identity of reviewers who chose to remain anonymous.

Reviewer 2 ·

Basic reporting

This is a well designed simulation study looking at evolution of asexual populations with varying levels of pleiotropy under Fisher's Geometrical Model.

Experimental design

The study is well conceived and explores the relevant parameter space.

Validity of the findings

The results of this study, though simple in scope, are consistent with experimental observations.

Reviewer 3 ·

Basic reporting

Review for: Competition and fixation of cohorts of adaptive 1 mutations under Fisher geometrical model (#9188)

Overall, I thought the MS was fine. The exploration of the FGM as a tool to explain clonal interference with varying degrees of pleiotropic mutations seems novel. I think researchers interested in theory to explain observations in evolve and resequence experiments for microbial populations may find these results useful.

--The figure axes need to be labeled and I would like the figure legends to be more descriptive. The figures and figure legends could be improved. For instance define variables in legends so that readers know what the parameters are if they just glance at the figures.

--Figure 4. I thought figure 4 was nice. I liked seeing the increase of multiple mutations.

--Line 36 – you state “We find that FGM under clonal interference, and with varying levels of pleiotropy, can reproduce the experimentally observed competition between different cohorts of mutations (“cohort interference”)”. I find the evidence connecting your theory to experimentally observed competition studies to be insufficient. This connection is key to your study being interesting and used. I think more emphasis on explicitly connecting your observation to experiments is needed.

--Line 66 – citations?? I would like a more thorough description of these experiments specifically how they relate to the clonal interference.

--Line 68-69 – wording of sentence a bit awkward. Not 100% sure what you mean.
--Define cohorts somewhere besides the abstract.

--I think you are making the assumption that many of your readers are aware of the experimental studies of clonal interference. I think the MS could be substantially improved by fleshing out the genetic expectations of clonal interference and specifically relating these to FGMs.

--Line 158 you state that “population with fewer traits reach higher fitness values”. This is only true within the time frame you looked – right? Seems that in Fig 1B the n=96 is still rising and hasn’t plateaued yet.

--I liked figure 2 and sup Fig. 21. Thought they both had good info. Only question I had is why were those ranges of n and m used?

--Figure 3: Axes labels? What about cases where m not equal to n?

--Figure 4: Axes labels? Were these just a random subset of your panels from the supplement Fig. 3? Did you carry these simulations out additional generations? It might be interesting to see a subset of these carried out more generations.

--discussion. I would like to see a clearer connection between the parameters you used (m and n) to experimental data. You mention some interesting studies whose results could be understood within the framework of FGM. I think the discussion could be better fleshed out connecting the observed results to experimental data.

Experimental design

--why did you choose m ranges 3..96 and n ranges of 3..96? What happens if n is much much larger than m? For instance say n=1000 and m=5? Seems that having n>>m might be biological relevant.

Validity of the findings

As presented, the results seem valid. I think a stronger emphasis on connecting theory to experiment, even in the discussion would make this paper stronger.

---

## Round 0.2 · accepted · Accept

· Academic Editor

Accept

Thank you for the very nice manuscript. We will be very pleased to publish this in PeerJ.

Reviewer 1 ·

Basic reporting

Authors did improve the manuscript in accordance with reviewers suggestions. Its writing style is clear. Though I do not dare to judge the English of the manuscript as I am not an English-speaker or professional editor.

Minor suggestions:
Line 179 in PDF of the manuscript: Is "Mutation cohorts of fixed in the initial steps of adaptation". Should it rather be "Mutation cohorts fixed in the initial steps of adaptation", without 'of'?

Line 305: Under FGM, cohort interference can be .... -> Under FGM, mutation cohort interference can be ....

Figures have been improved and are clear and easy to read. Including the new figures added after the review.

Experimental design

Experimental design is almost identical as in the first manuscript. Suggestions of the 3rd reviewer were implemented and improved the report.

Validity of the findings

No Comments

Reviewer 3 ·

Basic reporting

The revisions resulted in a significantly improved MS. As far as I can tell it is experimentally sound. The additional simulations and descriptions are helpful. I think this MS is fit for publication.

Experimental design

Solid and reasonable.

Validity of the findings

Findings are valid and interesting.